**Data Availability Statement:** All single nucleotide variants identified using whole exome sequencing in three affected family members are available in S3 Table.

# Prioritization of candidate genes for a South African family with Parkinson's disease using *in-silico* tools

Boiketlo Sebate[1◉], Katelyn Cuttler[1◉], Ruben Cloete[2]*, Marcell Britz[3], Alan Christoffels[2], Monique Williams[4¤], Jonathan Carr[5], Soraya Bardien[1]*

1 Division of Molecular Biology and Human Genetics, Department of Biomedical Sciences, Faculty of Medicine and Health Sciences, Stellenbosch University, Cape Town, South Africa, 2 South African Medical Research Council Bioinformatics Unit, South African National Bioinformatics Institute, University of the Western Cape, Cape Town, South Africa, 3 Greenacres Medical Centre, Port Elizabeth, South Africa, 4 Division of Molecular Biology and Human Genetics, Department of Biomedical Sciences, NRF/DST Centre of Excellence for Biomedical Tuberculosis Research, South African Medical Research Council Centre for Tuberculosis Research, Faculty of Medicine and Health Sciences, Stellenbosch University, Cape Town, South Africa, 5 Division of Neurology, Department of Medicine, Stellenbosch University, Cape Town, South Africa

◉ These authors contributed equally to this work.
¤ Current address: Department of Molecular and Cell Biology, University of Cape Town, Cape Town, South Africa
* sbardien@sun.ac.za (SB); ruben@sanbi.ac.za (RC)

## Abstract

Parkinson's disease (PD) is a neurodegenerative disorder exhibiting Mendelian inheritance in some families. Next-generation sequencing approaches, including whole exome sequencing (WES), have revolutionized the field of Mendelian disorders and have identified a number of PD genes. We recruited a South African family with autosomal dominant PD and used WES to identify a possible pathogenic mutation. After filtration and prioritization, we found five potential causative variants in *CFAP65*, *RTF1*, *NRXN2*, *TEP1* and *CCNF*. The variant in *NRXN2* was selected for further analysis based on consistent prediction of deleteriousness across computational tools, not being present in unaffected family members, ethnic-matched controls or public databases, and its expression in the substantia nigra. A protein model for NRNX2 was created which provided a three-dimensional (3D) structure that satisfied qualitative mean and global model quality assessment scores. Trajectory analysis showed destabilizing effects of the variant on protein structure, indicated by high flexibility of the LNS-6 domain adopting an extended conformation. We also found that the known substrate N-acetyl-D-glucosamine (NAG) contributed to restoration of the structural stability of mutant NRXN2. If *NRXN2* is indeed found to be the causal gene, this could reveal a new mechanism for the pathobiology of PD.

**Funding:** SB and JC received support from the National Research Foundation of South Africa (Grant Number: 106052) and the South African Medical Research Council (Self-Initiated Research Grant). RC and AC were supported by the South African Research Chairs Initiative of the Department of Science and Technology and National Research Foundation (NRF) of South Africa (award number UID 64751). The funders had no role in study design, data collection and analysis, decision to publish, or preparation of the manuscript.

**Competing interests:** The authors have declared that no competing interests exist.

# Introduction

Parkinson's disease (PD), the most common neurodegenerative movement disorder, is characterized by loss of dopaminergic neurons in the substantia nigra. PD affects motor, autonomic and cognitive functioning, as well as overall mood and behavior [1]. Research on PD has firmly established the role of genetic factors in its etiology. To date, several PD-associated genes have been identified through linkage analysis including *LRRK2*, *SNCA*, *DJ-1*, *PRKN*, *PINK1* and *SYNJ1* [2, 3]. More recently, next-generation sequencing (NGS) has contributed to the discovery of novel genes including *VPS35* [4, 5], *CHCHD2* [6], *VPS13C* [7], *TMEM230* [8] and *LRP10* [9]. These NGS technologies, namely whole exome sequencing (WES) and whole genome sequencing (WGS), provide the entire exome or genome of an individual at an affordable cost [10]. WES generates 20,000 to 30,000 variants, per individual [11], far less compared to WGS. However, it still requires filtration and prioritization analysis of variants into a workable list of candidates because it is not feasible to determine the effect of all variants, with functional experiments.

Currently, there is no standard filtration protocol, and it is a lengthy and complicated task that requires the use of several bioinformatic platforms [12]. These computational prediction tools allow prioritization of potentially deleterious variants over other variants. Sequence-based tools use computational algorithms based on amino acid physicochemical properties, protein structure and cross-species conservation. These include SIFT [13], PolyPhen-2 [14], MutationTaster [15], MAPP [16] and Panther [17]. Studies have compared multiple tools concluding that the algorithms were informative and valuable in determining the impact of the variant, although they had high rates of both false-positive and false-negative predictions [18, 19]. To address this shortfall, new meta-tools were developed such as CADD (Combined Annotation Dependent Depletion) [20, 21]. CADD achieves better performance by combining the predictive scores of multiple prediction tools into one unified score of potential deleteriousness, and for all 8.6 billion possible human mutations, it compares variants that survived natural selection with simulated mutations.

Assessing the frequency of genetic variants in population databases such as the Genome Aggregation Database (gnomAD) [22], the 1000 Genomes Project [23] and dbSNP [24] can also be used to prioritize variants. A minor allele frequency (MAF) threshold of $\leq 1\%$ is typically used to select potential pathogenic mutations by filtering out polymorphisms, based on the premise that risk alleles occur at low frequencies due to negative selection against deleterious mutations [25]. Furthermore, co-segregation analysis of the variant within families is invaluable for interpreting the variant's significance. However, phenocopies (present at relatively high levels in PD [26]) as well as non-manifesting mutation carriers need to be taken into account in family studies.

In the present study, a multiplex South African family with PD was recruited for genetic analysis. WES variants were filtered and prioritized using various computational tools, and additionally structural methods (DUET webserver) and molecular dynamic simulation analysis were performed. Using this approach, we excluded all known PD-associated mutations and identified a novel variant in a gene, not previously implicated in PD, for further functional studies.

# Materials and methods

## Ethical considerations

Ethical approval was obtained from the Health Research Ethics Committee of Stellenbosch University, Cape Town, South Africa (Protocol number 2002/C059). The research was

performed in accordance with the relevant guidelines and regulations. Written informed consent was obtained from all study participants.

## Study participants

Initially, we selected 11 multiplex South African families from our PD study collection for WES. These families were selected on the basis of having DNA available of at least two affected second degree relatives, PD had been diagnosed by a neurologist, and at least one individual had young-onset PD. WES was performed on a total of 24 affected individuals from these families. One of these families self-identified as Afrikaner and is the subject of the present study. The Afrikaner is a group that is unique to South Africa, and whose ancestry can be traced to people of predominantly Dutch but also German and French ancestry [27]. The family (ZA 253) comprises five PD affected individuals who were examined by a movement disorder specialist and diagnosed according to the UK Parkinson's Disease Society Brain Bank Research criteria [28]. A total of 218 unrelated ethnic-matched individuals were recruited as controls from the Western Province Blood Transfusion Service in Cape Town. These individuals were not clinically assessed for PD. Peripheral blood samples were collected from the study participants and genomic DNA was extracted according to the procedure for the phenol-chloroform DNA extraction, as previously described [29].

## Whole exome sequencing

WES was performed on three affected individuals using an Illumina HiSeq 2000, enrichment for exonic regions was performed according to the Agilent SureSelect Custom Enrichment Kit protocol. Burrows-Wheeler Aligner (BWA-MEM) [30] was used to align sequence reads or assembly contigs against a large reference, duplicate removal, SNP calling and indel detection. ANNOVAR software [31] was used to annotate the variants using GRCh37/human genome 19 as the reference genome. The Genome Analysis Toolkit (GATK) was used for variant calling and described variants as either exonic, intronic, in the UTR or splice-site region by comparison with a reference sequence [31]. To provide a summary of the coverage of mapped reads on a reference sequence at a single base pair resolution, SAMtools mpileup and the Minimum variant QUAL score of 30 was used.

## Validation using Sanger sequencing

Oligonucleotide primers were designed using sequence data obtained from the Ensembl Genome Browser database (http://www.ensembl.org). *In silico* verification of primers was conducted on Primer3 software version 4.0.0 (http://primer3.ut.ee) [32], as well as on Basic Local Alignment Search Tool (BLAST) (http://www.ncbi.nlm.nih.gov/BLAST). Forward and reverse PCR primers sequences, annealing temperatures and the sizes of the PCR products are available upon request.

## Polymerase chain reaction (PCR)

To amplify the fragments of interest for Sanger sequencing, 25µl PCR reactions containing template genomic DNA, 20 µmoles of each of the forward and reverse primers, 2.5 µM dNTPs (Promega, Madison, Wisconsin, USA), 1.5 nm MgCl2, 1X Green GoTaq® Reaction Buffer (Promega, Madison, Wisconsin, USA) and 0.01U GoTaq® G2 Flexi DNA Polymerase (Promega, Madison, Wisconsin, USA) was used. Amplification was performed in an ABI 2720 Thermal Cycler (Applied Biosystems Inc., Foster City, California, USA). To visualize the PCR amplicons and to investigate if non-specific primer binding or contamination was present,

agarose gel electrophoresis was used. PCR amplicons were visualized using a SynGene UV gel documentation system (Synoptics Ltd., Cambridge, UK) with GeneTools software version 3.0.6 (Synoptics Ltd., Cambridge, UK).

## Screening of controls

HRM analysis detects the shift in fluorescence as a double-stranded PCR product dissociates to single-stranded DNA with increasing temperature. For this analysis, PCR was performed with the inclusion of 1% SYTO9 fluorescent dye (Invitrogen, USA). The melting temperature conditions ranged from 75°C to 95°C rising by 0.1°C increments on a Rotor-Gene 6000 analyzer (Corbett Life Science, Australia). The resulting thermal denaturation profile is unique to that specific PCR product, because DNA strand melting depends on sequence length, bases and GC content (HRM Assay Design and Analysis Booklet; http://www.corbettlifescience. com/shared/Rotor-Gene%206000/hrm_corprotocol.pdf). Samples with known variants were included as positive controls, for comparison. A negative control was included in each run to monitor contamination.

## *In-silico* functional prediction tools

To investigate the variants identified, computational analysis was performed. Public mutation databases were searched to determine the frequency of the variants, these included EXAC Database (http://exac.broadinstitute.org/), gnomAD (https://gnomad.broadinstitute.org/), the 1000 Genomes Project (http://browser.1000genomes.org/index.html) and dbSNP (https:// www.ncbi.nlm.nih.gov/snp). Functional predictions were determined from sequence homology–based programs namely Sorting Intolerant From Tolerant (SIFT; http://sift.jcvi.org/), PolyPhen-2 (http://genetics.bwh.harvard.edu/pph2/), MutationTaster (http://www. mutationtaster.org/) and Combined Annotation Dependent Depletion (CADD; http://cadd. gs.washington.edu/). To determine evolutionary constraint acting on genomic sites GERP+ + was used (http://mendel.stanford.edu/SidowLab/downloads/gerp/).

SIFT and PolyPhen-2 both report results in terms of pathogenic scores accompanied by the prediction (Benign, Tolerated or Deleterious). MutationTaster scores the amino acid change then reports one of two predictions based on its disease-causing threshold (Disease causing or Neutral). CADD integrates 63 predictive features in total which includes the scores of SIFT, GERP++, PolyPhen-2 as well as CpG distance (a short stretch of DNA in which the frequency of the CG sequence is higher than other regions), and total GC content. It reports phred-like scores ("scaled C-scores") ranging from zero to 50, the variant that is predicted to be among the 10% most deleterious substitutions that can occur in the human genome at that specific base position, is assigned a score of 10 or greater. While variants in the 1% most deleterious substitutions are assigned values of 20 or greater and those within 0.1% of the highest possible substitutions at that specific locus are assigned values of 30 or greater. Thus, the higher the CADD score the more likely it is that the variant is highly pathogenic. GERP++ uses maximum likelihood evolutionary rate estimation for position-specific scoring, the score ranges from -12.3 to 6.17. The closer the score is to 6.17, a value that represents the most conserved a region can be, the greater the level of evolutionary constraint inferred to be acting on that site.

## Pathway and expression analysis

To prioritize the variants, pathway analysis was performed using KEGG Pathways Analysis (http://www.genome.jp/kegg/pathway.html) and Pathway Analysis (http://www.pantherdb. org/pathway/).The publicly available expression databases accessed were the Allen Brain Atlas (www.brain-map.org/) and Human Protein Atlas (https://www.proteinatlas.org/). Both

database record human mRNA expression data but Allen brain Atlas gives only brain regional whole-transcriptome gene expression data.

## NRXN2 protein structure modelling

The three dimensional (3D) structure for human NRXN2 has not yet been resolved experimentally and was predicted using the online Swissmodel Webserver [33]. Prior to modelling, Swissmodel constructs an alignment between the target amino acid sequence of NRXN2 and potential homologous templates by performing a position specific iterative (PSI-BLAST) against the non-redundant protein databank (PDB) sequences [34]. Once a suitable template was identified then the actual modelling step was invoked by submitting the alignment to the Swissmodel webserver. Subsequent, to the modelling of NRXN2 the quality of the protein model was assessed using Swissmodel inbuilt assessment methods Global Model Quality Estimation (GMQE) and QMEAN6 composite score as well as performing a structural alignment of atoms between the NRXN2 protein structure and the homologous template using the Pymol align command to calculate the RMSD [35]. Usually models with a lower RMSD value suggest very little deviation between the backbone atoms of the protein model and the homologous template.

## Molecular dynamic simulation

We prepared in total four simulation systems for NRXN2, each consisting of the wild type (WT) NRXN2 with and without sugar moeity N-acetyl-D-glucosamine (NAG) and mutant (MUT) structures NRNX2 with and without NAG, for simulation studies. The sugar moiety NAG was extracted from the homologous template PDBID: 3R05 (NEUREXIN-1-ALPHA; BOS TAURUS) by superimposition using PyMol. The NAG was included because it is important for glycosylation of membrane proteins whereby glycans are attached to proteins necessary for protein-protein interactions. All the simulations were carried out using the GROMACS-5.1 package [36] along with the CHARMM36 all-atom force field [37]. The accurate topologies for NAG was generated using SwissParam tool [38]. All four systems were solvated with TIP3 water molecules in a cubic box of at least 18 Å of water between the protein and edges of the box. To neutralize the negative charge of the WT and MUT systems without NAG, WT and variant with NAG systems, 14, 15, 14 and 15 sodium ions were added to each system, respectively.

Each system underwent 50,000 steps of steepest descents energy minimization to remove close van der Waals force contacts. Subsequently, all systems were subjected to a two-step equilibration phase namely; NVT (constant number of particles, Volume and Temperature) for 500 ps to stabilize the temperature of the system and a short position restraint NPT (constant number of particles, Pressure and Temperature) for 500 ps to stabilize the pressure of the system by relaxing the system and keeping the protein restrained. For the NVT simulation the system was gradually heated by switching on the water bath and the V-rescale temperature-coupling method [39] was used, with constant coupling of 0.1 ps at 300 K under a random sampling seed. While for NPT the Parrinello-Rahman pressure coupling [40] was turned on with constant coupling of 0.1 ps at 300 K under conditions of position restraints (all-bonds). For both NVT and NPT electrostatic forces were calculated using Particle Mesh Ewald method [41]. All systems were subjected to a full 100 ns simulation and these were repeated twice for 100 ns to validate reproducibility of results. The analysis of the trajectory files was done using GROMACS utilities. The root mean square deviation (RMSD) for backbone atoms was calculated using gmx rmsd, RMSF average per-residue analysis using gmx rmsf. The change in the solvent accessibility surface area (SASA) for protein atoms was calculated using gmx sasa and

the radius of gyration for the backbone atoms was calculated using gmx gyrate to determine if the system reached stability and compactness over the 100 ns simulation. VMD [42] was used to visually inspect changes in secondary structural elements and motion of domains along the trajectory.

## Principal component analysis

Principal component analysis (PCA) is a statistical technique that reduces the complexity of a data set and was used here to extract biologically relevant movements of protein domains from irrelevant localized motions of atoms. For PCA analysis the translational and rotational movements was removed from the system using g_covar from GROMACS to construct a covariance matrix. Next, the eigenvectors and eigenvalues were calculated by diagonalizing the matrix. The eigenvectors that correspond to the largest eigenvalues are called "principal components", as they represent the largest-amplitude collective motions. We filtered the original trajectory and project out the part along the most important eigenvectors namely: vector 1 and 2 using g_anaeig from GROMACS utilities. Furthermore, we visualized the sampled conformations in the subspace along the first two eigenvectors using g_anaeig in a two-dimensional projection. The two-dimensional projection of the first two principal components was plotted using Gnuplot version 4.4 [43]. Afterwards, we calculated the free energy surface (FES) using the program g_sham and plotted it using xpm2mat.py script and Gnuplot in a 3n matrix along the two order parameters, Rg and RMSD. The FES represents all the possible different conformations a protein can adopt during a simulation and are typically reported as Gibbs free energy. The molecules free energy was calculated with the formula $\Delta G(r) = -k_BT$ in $P(x,y)/P_{min}$, where P is the probability distribution of the two variables Rg and RMSD, Pmin is the maximum probability density function, kB is the Boltzmann constant and T is the temperature of the simulation. Conformations sampled during the simulation are projected on a two dimensional plane to visualize the reduced free energy surface. The clustering of points in a specific cell represents a possible metastable conformation. All simulations were carried out using the GROMCS-5.1 package [36] along with the CHARMM36 all-atom force field [37].

## NRXN2 variant structure stability predictions

The mCSM webserver was used to assess the effect of the mutation on the stability of the protein structure of NRNX2 [44]. First, we extracted six structures (every 10ns) in total over the last 50ns of the simulation trajectory. Next, each WT structure without NAG was uploaded to the webserver and the position of the variant was specified as G889D on the webpage. mCSM calculates a Delta-delta G stability score and provides a phenotypic assessment of the score with destabilizing scores being negative, while stabilizing scores are positive [45].

## Results

### Description of the family

The complete pedigree of this family, which is denoted as family ZA253, contains 57 individuals, and in total, eight members were reported to have PD. A simplified version of the pedigree indicating the family members, who took part in this study, is shown in Fig 1. The patient labelled as individual III-8 (proband) was the first individual to be diagnosed with PD and his age at onset (AAO) was 48 years. He initially presented with anxiety and difficulty with the use of his leg following a general anesthetic. He later developed moderate to high amplitude rest tremor. Levodopa improved his condition markedly, but he developed dystonia of the left leg within weeks of starting medication. His siblings were subsequently assessed, and his brother

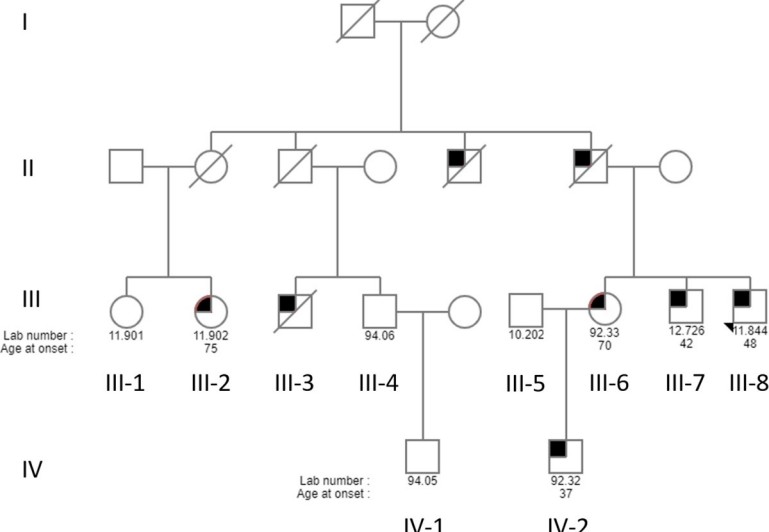

**Fig 1. Pedigree of the South African family ZA253.** For readability and confidentiality, the pedigree is greatly simplified. Circles denote females and squares depict males. The filled in symbols indicate affected individuals. The diagonal line indicates that the person is deceased, and the arrow indicates the proband. The numbers below each individual is the laboratory ID number and age at onset of disease. Branches without clinically confirmed PD or without DNA samples were omitted, but all known living family members with PD diagnoses are included.

(III-7) was also confirmed as having PD. His AAO was 42 years, at which time he developed difficulties with concentration together with weakness of the legs and gait disturbances. On examination, he had marked dystonia of the left arm, and very asymmetrical bradykinesia and rigidity. Patient IV-2, the nephew of the two affected siblings, was diagnosed with early-onset PD at the age of 37 years. Examination revealed facial dystonia, rigidity at the right wrist, mild generalized bradykinesia, and slowness of gait [Unified Parkinson's Disease Rating Scale (UPDRS) motor score 16]. Furthermore, his mother (individual III-6) was examined at the age of 67 years; at that time her examination was normal. Repeat examination three years later (aged 70 years) showed mild generalized bradykinesia (UPDRS motor score 11), although she had no complaints. Individual III-2 was the most recent individual in the family to be diagnosed with PD, in 2017. However, she presented as typical idiopathic PD, with AAO of 75 years. The symptoms at the time of diagnosis were tremor and rapid eye movement sleep behavior disorder, with no evidence of dystonia. A summary of the family's clinical characteristics is provided in S1 Table.

## Whole exome sequencing

WES was performed on three of the affected family members (III-7, III-8 and IV-2). These three patients all appear to have young onset PD, with prominent dystonia, resembling similar cases with *PRKN* and *PINK1* mutations. Moreover, all three cases are characterized by a long duration of illness, mild autonomic impairment, and delayed and mild impairment of cognition, with absence of REM sleep behavior disorder. DNA of individual III-2 was not available at the time that the WES was performed.

The WES metrics (S2 Table), revealed good overall coverage of the target regions in all three individuals. For the data analysis, we assumed that the three affected individuals have the same genetic cause of disease and that the mode of inheritance is autosomal dominant. Importantly, none of the known PD mutations were found in any of the affected individuals which suggests that there is likely a novel PD-causing mutation in this family.

**Table 1. Rare and novel exonic missense variants shared between the three affected individuals after variant filtering.**

| | | CFAP65 | RFT1 | NRXN2 | TEP1 | CCNF |
|---|---|---|---|---|---|---|
| | | p.T1023A | p.A463G | p.G849D | p.Y412C | p.C363S |
| Affected individuals | III-8 (proband) | **Present** | **Present** | **Present** | **Present** | **Present** |
| | III-7 | **Present** | **Present** | **Present** | **Present** | **Present** |
| | IV-2 | **Present** | **Present** | **Present** | **Present** | **Present** |
| | III-6 | **Present** | **Present** | **Present** | **Present** | **Present** |
| | III-2 | **Present** | Absent | Absent | Absent | Absent |
| Unaffected individuals | III-4 | Absent | Absent | Absent | Absent | **Present** |
| | IV-1 | Absent | Absent | Absent | Absent | **Present** |
| | III-1 | Absent | Absent | Absent | Absent | **Present** |
| | III-5 | Absent | Absent | Absent | Absent | Absent |

Present, variant present; Absent, variant absent

After exclusion of synonymous, common, non-co-segregating and homozygous variants, a total of nine novel or rare non-synonymous heterozygous variants were shared between the three affected individuals. These variants are in nine genes, *ACTN3* (Actinin Alpha 3), *CCNF* (Cyclin F), *CDC27* (Cell Division Cycle 27), *CFAP65* (Cilia and Flagella Associated Protein 65), *RFT1* (Requiring Fifty-Three 1), *NRXN2* (Neurexin 2), *POU2F1* (POU Class 1 Homeobox 1), *TEP1* (Telomerase Associated Protein 1) and *TUBB6* (Tubulin Beta 6 Class V).

## Sanger sequencing of variants and co-segregation in family

Sanger sequencing validation was performed to ensure that the variants are not NGS artefacts, and five variants in *CCNF*, *CFAP65*, *RFT1*, *NRXN2* and *TEP1* were validated. Thereafter, we performed genotyping of the family and these results are summarized in Table 1. The most recently diagnosed patient III-2 had only one of the five validated variants, p.T1023A in *CFAP65*. The unaffected family members shared only one variant, p.C363S in *CCNF*, with the affected individuals, which indicates that it may be a polymorphism.

## Analysis using functional prediction tools

Various functional prediction tools were used to identify and prioritize variants that are predicted to have a major impact on the protein. Notably, none of the codon changes occurred in the interchangeable third base position (wobble base). The results are summarized in Table 2 with the functional scores and predictions from SIFT, PolyPhen-2, MutationTaster, GERP++,

**Table 2. *In-silico* functional tool scores and predictions for the five variants validated with Sanger sequencing.**

| Gene | cDNA position | Amino acid and codon change | SIFT prediction | PolyPhen-2 prediction | Mutation Taster prediction | GERP prediction | CADD prediction | Condel prediction |
|---|---|---|---|---|---|---|---|---|
| *CFAP65* | c.A3151G | p.T1023A **A**CC⇒**G**CC | 0; Deleterious | 0.124; Benign | 1; Disease causing | 2,38; Conserved | 18,70; Deleterious | 0.763 Deleterious |
| *RFT1* | c.C1450G | p.A463G G**C**T⇒G**G**T | 0.49; Tolerated | 0.001; Benign | 0; Neutral | 2,53; Conserved | 4,96; Benign | 0.042 Neutral |
| *NRXN2* | c.G3008A | p.G849D G**G**C⇒G**A**C | 0; Deleterious | 0.973; Probably damaging | 1; Disease causing | 4,93; Conserved | 29,50; Deleterious | 0.831 Deleterious |
| *TEP1* | c.A1276G | p.Y412C T**A**C⇒T**G**C | 0.51; Tolerated | 0; Benign | 0; Neutral | -8,29; Not conserved | 0,00; Benign | 0.050 Neutral |
| *CCNF* | c.G1176C | p.C363S T**G**C⇒T**C**C | 0.06; Tolerated | 0.939; Probably damaging | 1; Disease causing | 5,43; Conserved | 27,30; Deleterious | 0.707 Deleterious |

CADD and Condel. The TEP1 p.Y412C variant was predicted to be benign across all of the different prediction tools with a CADD score of zero and with a GERP++ score of—8,29. The conservation score implies that in the region of the genome where this variant occurs, there are more substitutions than the average neutral site and thus indicates that this region may not be under evolutionary constraint. The p.A463G variant in RFT1 was also predicted to be benign by all of the tools though its conservation score (of 2,53) implies that it occurs in a region that is evolutionarily-conserved. The p.T1023A variant in CFAP65 and the p.C363S variant in CCNF are predicted to be pathogenic by most of the tools, and both have positive scores indicating that they are found in conserved regions of the genome. The only variant that was predicted to be pathogenic across all of the functional prediction tools was p.G849D in NRXN2, which had a CADD score of 29,50 and a conservation score of 4,93 showing that the level of evolutionary constraint inferred to be acting on this position is relatively high.

## Frequency of variants in control populations

To determine the frequency of the five variants in controls, a search was performed of publicly available databases including the 1000 Genomes Project, dbSNP and gnomAD. This revealed that four of the variants are either extremely rare or are novel according to data available on African American, American, East Asian, South Asian, Finnish and Non-Finnish European populations. CFAP65 p.T1023A was found in gnomAD with a MAF frequency of 8.88e-5 (25 out of 281690 alleles) (https://gnomad.broadinstitute.org/variant/2-219886565-T-C?dataset= gnomad_r2_1).

When assessing the frequency of these variants, it is important to consider not just the publicly available online databases but also to screen ethnically matched controls. Therefore, we screened South African ethnic-matched controls, but none of the variants was present in these individuals. The *TEP1* p.Y412C, *RFT1* p.A463G and *CCNF* p.C363S variants were screened in 192 individuals, and the *CFAP65* p.T1023A and *NRXN2* p.G849D variants were screened in 218 and 216 individuals, respectively.

## Pathway and expression analysis

To add further information about the possible causal role of the variants, gene expression profiles were assessed using publicly available databases, namely the Allen Brain Atlas [46] and the Human Protein Atlas [47]. Pathway analysis was also performed using KEGG Pathways Analysis [48] and Panther Pathway Analysis [49] to determine if any of the five candidates are co-expressed, co-regulated or co- localize with each other or any of the known PD genes. This analysis also shows if the variants and any of the known PD genes are functionally related or impact a pathway of interest or even a trait of interest such as neuronal development, regulation and functioning. The results show that none of the five proteins are co-expressed, co-regulated or co-localize with each other or any of the proteins encoded by the known PD genes. Notably, NRXN2 was found to be highly expressed in the brain including the substantia nigra (https://www.proteinatlas.org/ENSG00000110076-NRXN/tissue). This is the region of the brain predominantly involved in PD pathogenesis and also thought to be involved in pathways that regulate synaptic functioning, neurotransmitter secretion and neuronal cell-to-cell adhesion. RFT1 is also expressed in the brain, however not particularly defined to any specific region. RTF1 was implicated in pathways of protein metabolism and the endoplasmic reticulum membrane network. CFAP65 was associated with the motile cilia pathway, TEP1 is reported to be involved in apoptotic pathways and in assembling telomerase components in cell signaling and CCNF was associated with pathways of ubiquitination and cell cycle regulation.

## Selection of NRXN2

After the above analyses, we prioritized one candidate gene (*NRXN2*) for further study. As mentioned previously, even though the *CCNF* p.C363S variant was predicted to be deleterious by most of the prediction tools, it was excluded at this stage as it was found in unaffected family members. The variants in *TEP1* (p.Y412C) and *RTF1* (p.A463G) were predicted to be benign across all the prediction tools used, therefore they were also excluded. Although the p.T1023A variant in *CFAP65* was predicted to be deleterious by the majority of the prediction tools, it functions specifically in ciliac processes, and thus is unlikely to be involved in PD pathobiology. Importantly, while affected individual III-2 has the *CFAP65* variant, we suspect that she is a phenocopy. Evidence for this is that her phenotype is typical late-onset PD that is unlike the other affected family members. Consequently, the p.G849D variant in *NRXN2* is considered to be the strongest candidate of the five variants. All of the computational algorithms predicted the G849D substitution as being potentially deleterious. The position where this variant occurs was reported to be highly conserved by GERP++ and it resulted in a codon change from a small, non-polar amino acid to a larger, negatively charged, polar one. None of the controls obtained from the online databases or in the local population had the variant. The *NRXN2* variant was also not present in exomes of 600 probands of French, North African and Turkish origins with predominantly early-onset PD (Suzanne Lesage, personal communication). NRXN2 was reported by two expression databases to be expressed in the brain and specifically in the substantia nigra, a key brain region involved in PD pathogenesis. NRXN2 was therefore selected for further studies involving protein structure modelling and simulation analysis.

## NRXN2 protein model

The Swissmodel webserver [33] search, identified homologous template PDBID: 3R05 crystal structure of neurexin 1 alpha [Laminin neurexin sex hormone binding globulin domains (LNS1-LNS6), with splice insert SS3, Bos taurus (cow)] as a highly similar protein to human NRNX2. The template 3R05 was chosen to construct a three-dimensional (3D) structure for NRXN2 because it shared 71% sequence identity and over 50% structural similarity with our target protein. The 3D structure built for human NRXN2 is structurally similar to template 3R05 displaying the five LNS2-6 and two EGF-like repeats excluding LNS domain 1 and one EGF-like domain (Fig 2). The protein model predicted for human NRXN2 had a GMQE score of 0.51 [value between 0 and 1] and a QMEAN Z-score of -1.67 that are close to zero and higher than -4, suggesting high reliability in the quality of the predicted protein model. Superimposition of the predicted model onto the homologous template structure indicated a root mean square deviation (RMSD) value of 0.226Å, which is less than 2Å, suggesting very little deviation in the main chain backbone atoms of the protein model and the template. Furthermore, the Glycine residue (G849) in the template and homology model is conserved and has a positive phi and psi dihedral angle conformation and therefore any amino acid substitution at this position will have a significant effect on the protein structure.

## NRXN2 molecular dynamic simulation analysis and stability predictions

Four simulation systems were prepared for NRXN2 analysis. These are wild type (WT) NRXN2 with and without the sugar moiety, N-acetyl-D-glucosamine (NAG), and mutant (MUT) NRXN2 with and without NAG. NAG was used in this analysis as it is important for the glycosylation of NRXN2 and it was extracted from the homologous template. The kinetic energy and thermodynamic properties were found to fluctuate around stable values with the potential and total energies values fluctuating at negative $1 \times 10^{-6}$ values and the temperature fluctuating around 300K for the four NRXN2 systems (S1–S3 Figs).

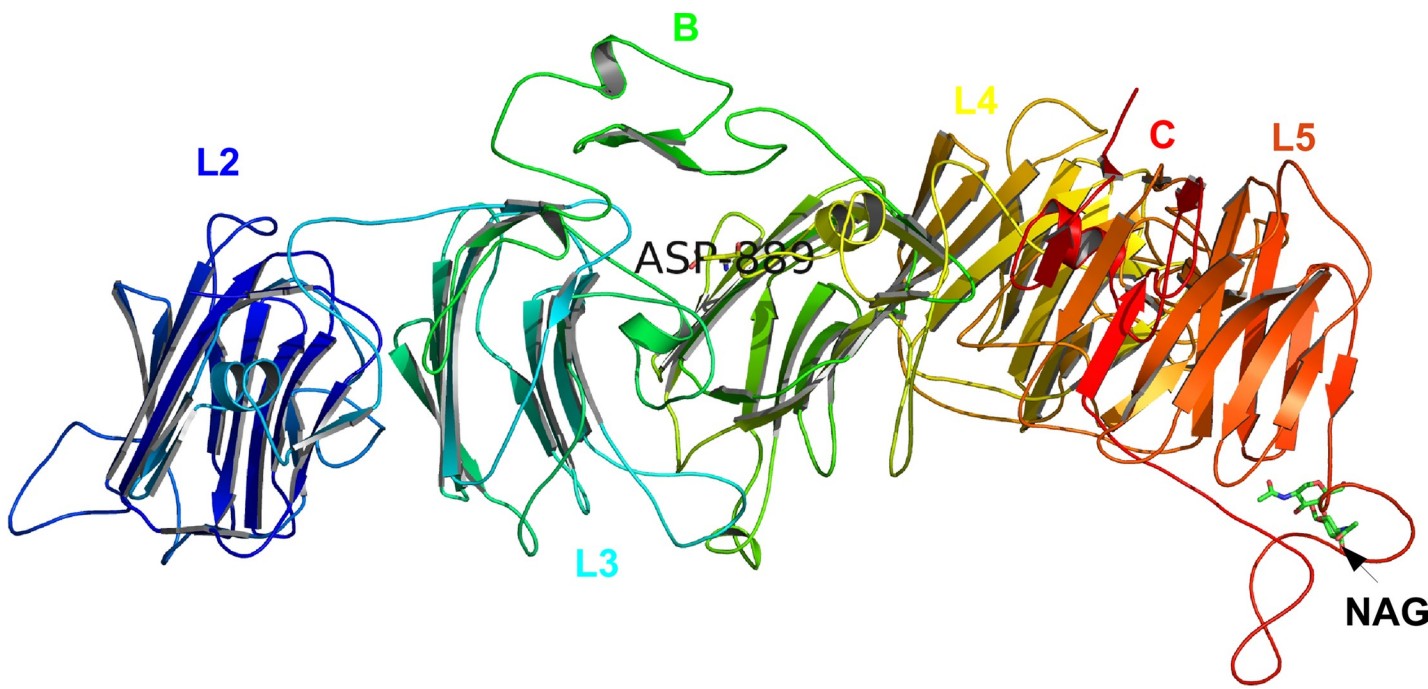

**Fig 2. Swissmodel predicted 3-dimensional (3D) structure for human NRXN2 with mutation p.G849D (p.G889D) in complex with N-acetyl-D-glucosamine (NAG).** Each domain is colour coded, EGF-like domains are labelled B and C, substrate NAG are labelled as well as the mutation, p.G889D in our protein model.

Measuring the mean and standard deviation of the RMSD values for the four systems indicated lower values of 1.61 nm ± 0.26 for the WT system without NAG compared to the MUT system without NAG of 2.16 nm ± 0.60 (Fig 3A). However, for the WT system with NAG, a higher RMSD value (2.25 nm ± 0.52) was observed compared to the MUT system with NAG (1.54 nm ± 0.55) (Fig 3A). The RMSD showed that the MUT systems with and without NAG reached equilibrium after 50ns, therefore subsequent analysis considered only the last 50ns of the simulation trajectories. The average root mean square fluctuation (RMSF) per-residue values for the four systems indicated lower fluctuation values for the WT system without NAG (0.32 nm ± 0.14) compared to the MUT system without NAG (0.53 nm ± 0.24) (Fig 3B). Furthermore, the RMSF fluctuation per-residue values for the other two systems with NAG indicated slightly lower values of 0.37 nm ± 0.18 for the WT system compared to 0.40 nm ± 0.16 for the MUT system (Fig 3B). Similarly, the radius of gyration values and the solvent accessible surface area values for the WT NRXN2 without NAG were significantly lower than that of the other three systems. The calculated Rg values for the backbone atoms for WT NRXN2 without NAG was lower (3.90 nm ± 0.03) compared to MUT NRXN2 without NAG, WT and MUT NRXN2 with NAG, (5.01 nm ± 0.10; 4.76 nm ± 0.11; 5.13 nm ± 0.08, respectively) (Fig 3C). The solvent accessible surface area values of the protein for WT and MUT NRXN2 in absence of NAG were lower (480.84 nm ± 5.43; 490.59 nm ± 4.46) compared to the WT and MUT NRXN2 in the presence of NAG (507.63 nm ± 4.26; 495.76 nm ± 4.78) (Fig 3D).

Calculation of the contribution of each of the top ten principal components (PCs) indicated that the first two PCs contributed significantly to the movement of the protein. For each system, PC1 contributed to 70%, 46%, 60% and 37% to WT no NAG, WT with NAG, MUT no NAG, and MUT with NAG for NRXN2, respectively. While, PC2 contributed 8%, 19%, 15% and 29% to WT no NAG, WT with NAG, MUT no NAG, and MUT with NAG for NRXN2, respectively. Therefore, 2D projections of the first and second principal components for all

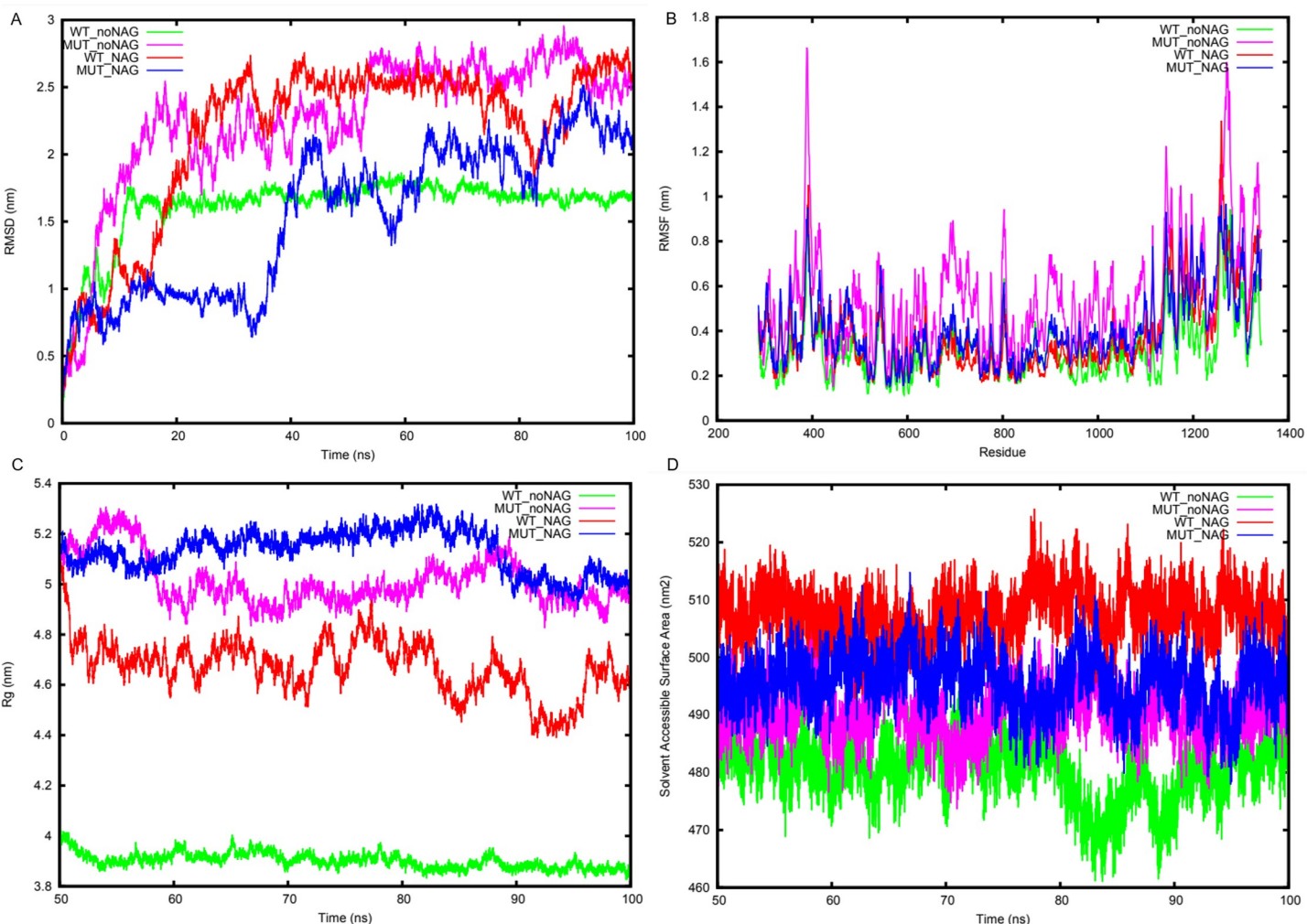

**Fig 3. Trajectory analysis of wild type (WT) NRNX2 and mutant (MUT) NRNX2, both with and without NAG. (A)** root mean square deviation (RMSD) deviation of the backbone atoms. MEAN + STDEV (1.61mm ± 0.26, 2.26mm ± 0.52, 2.16mm ± 0.60 and 1.54mm ± 0.55). **(B)** The average RMSF fluctuation per-residue. MEAN + STDEV (0.31mm ± 0.14, 0.54mm ± 0.24, 0.37mm ± 0.18 and 0.40mm ± 0.19). **(C)** Radius of gyration of the backbone atoms. MEAN + STDEV (3.90mm ± 0.03, 5.01mm ± 0.10, 4.67mm ± 0.11 and 5.13mm ± 0.08). **(D)** Solvent accessible surface area of the protein. MEAN + STDEV (3.90mm ± 0.03, 5.01mm ± 0.10, 4.67mm ± 0.11 and 5.13mm ± 0.08). Line colors: WT_noNAG = green, MUT_noNAG = light magenta, WT_NAG = red and MUT_NAG = blue.

four systems were plotted and shown in Fig 4. Calculation of the covariance matrix values after diagonalization showed a significant decrease for the WT NRXN2 without NAG (237.82 nm) system compared to the other three systems MUT without NAG, WT and MUT with NAG (612.38 nm; 508.19 nm; 554.72 nm, respectively) (Fig 4).

We performed interaction analysis to determine which residues played a role in the binding of NAG to the protein in the WT NRXN2 and identified one conserved hydrogen bond interaction with Arg1266 (in our model) as an important anchor point that could be exploited for drug design (S4 Fig). The first repeat of the four simulation systems showed similar results to the first run for thermodynamic and kinetic energy parameters (S5–S7 Figs). Additionally, equilibrium is reached after 50ns of the simulation run and the WT system without NAG had higher stability values based on RMSD analysis compared to the MUT system without NAG (S8 Fig). Similarly, the repeat 2 showed convergence of energy and temperature terms (S9–S11

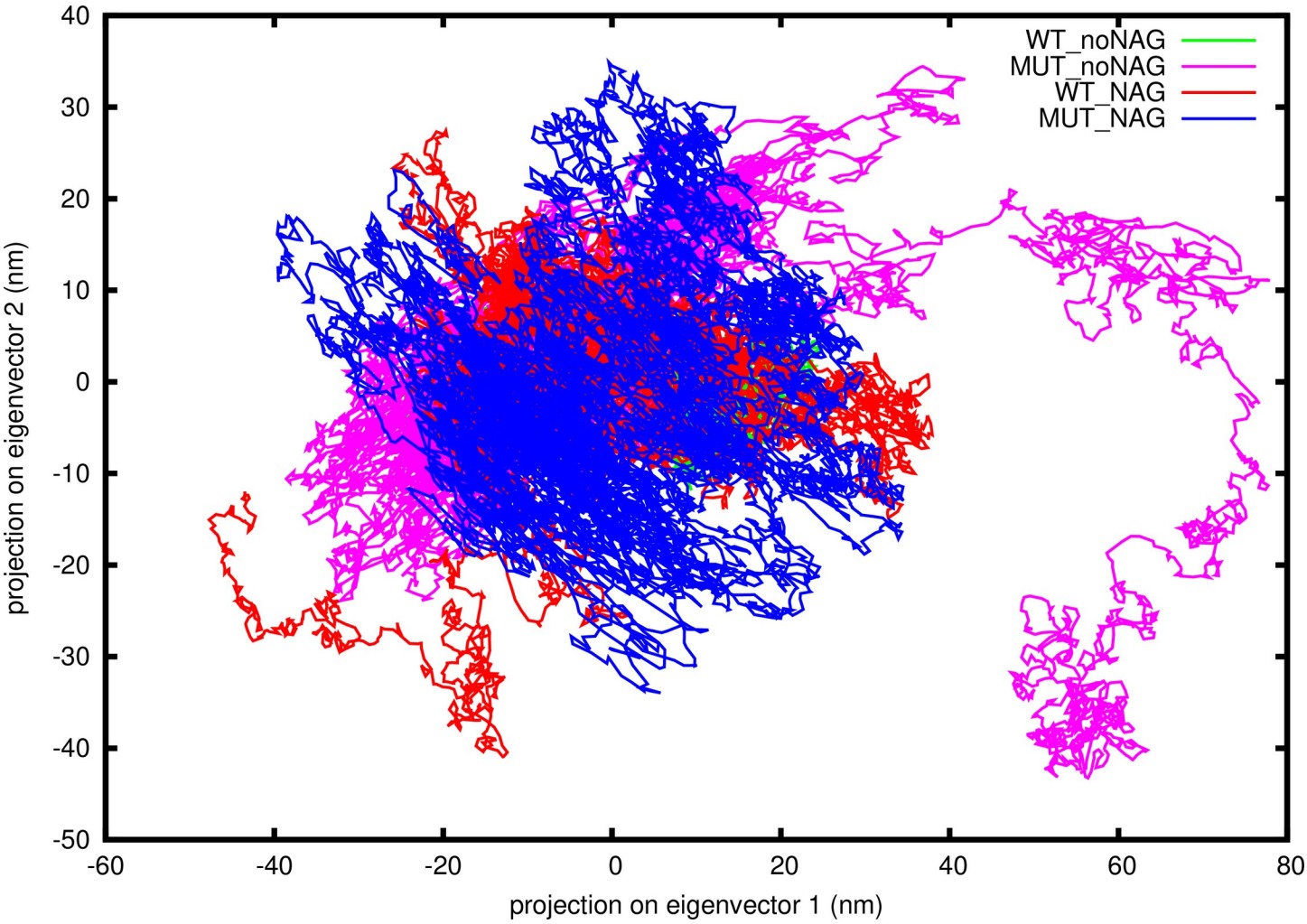

**Fig 4. Two-dimensional (2D) projections of the first and second principal components for the WT NRNX2, MUT NRNX2 without NAG and WT NRNX2, MUT NRNX2 with NAG systems.** Covariance matrix after diagonalization values for each system; 237.82 nm, 612.38 nm, 508.19 nm and 554.72 nm. Line colors: WT_noNAG = green, MUT_noNAG = light magenta, WT_NAG = red and MUT_NAG = blue.

Figs), this is in agreement with repeat 1, while the RMSD analysis for repeat 2 confirmed stability values of repeat 1 (S12 Fig).

The change in secondary structure for the NRNX2 WT and MUT without NAG is visually shown in a short simulation movie over the last 50ns for the simulation trajectory (S1 and S2 File). In these movies, the LNS6 domain undergoes large transitions between flexed and extended conformations for the mutant structure without NAG while the LNS6 domain for the WT without NAG remains in a stable flexed conformation. FES analysis of the four systems identified several (up to 6) metastable conformations for the WT system without NAG, while the MUT without NAG showed only one energy minima state. In contrast, the other two systems with NAG adopted at least one metastable state for the MUT structure and none for the WT structure (Fig 5A–5D). The WT system without NAG seems to be more stable than the MUT without NAG due to the six energy minima states. The opposite is found for the systems with NAG, as the MUT with NAG becomes more stable while the WT with NAG becomes more flexible. Furthermore, stability predictions using the mCSM webserver was performed to determine the effect of the novel variant p.G849D (in our model p.G889D) on the NRXN2

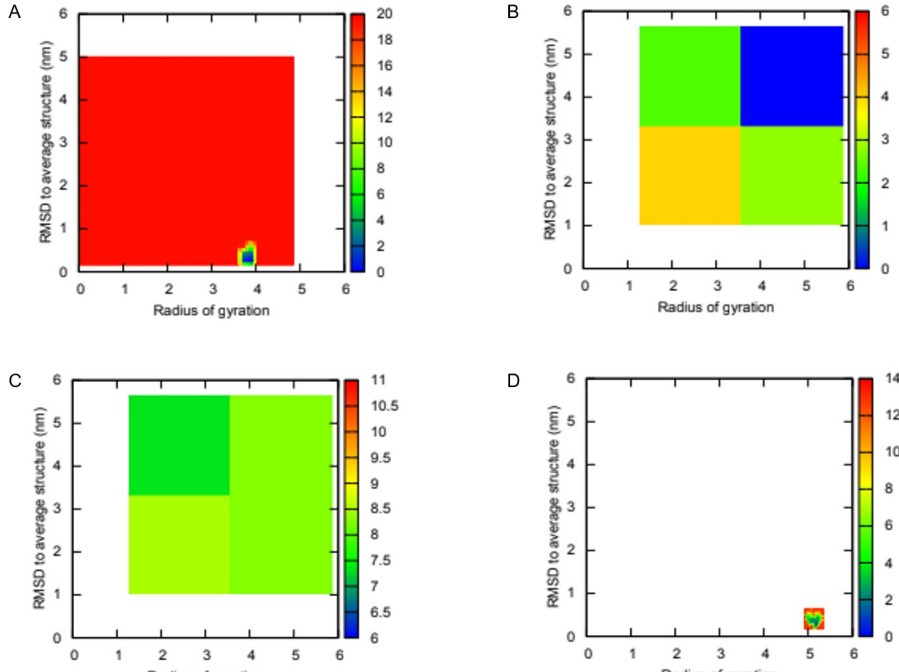

**Fig 5. Two-dimensional (2D) free energy landscapes plotted for NRXN2 along two order parameters root mean square deviation (RMSD) to average structure and Rg.** (**A**) WT without NAG (**B**) MUT without NAG (**C**) WT with NAG (**D**) MUT with NAG. Blue regions represent low energy conformational states while red indicate high energy states.

protein structure. The results were in concordance for all six structures extracted over the last 50 ns indicating destabilizing delta-delta ($\Delta\Delta G$) scores of -1.389, -1.131, -1.573, -1.183, -1.298 and -1.596 for 50, 60, 70, 80, 90 and 100 ns frames, respectively.

## Discussion

The absence of known PD mutations in this South African family led to the conclusion that a novel mutation could be responsible for the disease phenotype. Four variants in *CFAP65*, *RFT1*, *NRXN2* and *TEP1* are shared by four of the affected family members and are not in the unaffected family members or the controls. It is possible that the variants in any of these genes or even in another gene is responsible for the disease in this family, however, at this stage, the *NRXN2* p.G849D variant was selected for further study. It is the top candidate even though patient III-2 does not share this variant, as this patient may be a phenocopy, known to occur in PD families [26, 50], since she has typical, late-onset sporadic PD unlike the other four affected individuals. This highlights the complexity of interpreting familial co-segregation in disorders where the genetic disease can be indistinguishable from idiopathic forms.

The *NRXN2* p.G849D change is from glycine a small, non-polar, side chain free amino acid, to aspartic acid a larger, negatively charged amino acid with a carboxyl group side chain. The 3D structure predicted for NRXN2 provided deeper insight into the structural implications of G849D on NRXN2 protein structure. The overall structure of NRXN2 is similar to that of the homologous template 3r05, and adopts a similar fold. Using molecular dynamic simulations to understand the overall movement of the protein structure added support to the destabilizing effect of the variant p.G849D on the mobility of the LNS6 domain (residues 1137–1343 in our model). Trajectory analysis confirmed the destabilizing effect of the variant

on the protein structure and the predicted increase in the flexibility of the LNS6 domain might prevent the binding of neuroligins and synaptic organization. Additionally, we showed that adding NAG to the mutant 'system' restored some stability to the NRXN2 protein. From these findings, we hypothesize that the novel variant p.G849D (in our model p.G889D) is destabilizing to NRXN2's protein structure, and that adding NAG to the mutated structure might reduce flexibility and restore its ability to interact with its known interactors such as neuroligin.

NRXN2 has been linked to pathways associated with neuronal and synaptic functioning, and expression analysis showed that it is expressed in the pars reticulata of the substantia nigra, the main region of the brain affected in PD. Its predicted functions range from neuronal cell-cell adhesion, neurotransmitter secretion regulation to synaptic regulation. Indeed, NRXN2 is known to mediate synaptic organization and differentiation [51, 52]. Recently, Naito et al. [51] investigated whether neurexins plays a role in Alzheimer's disease (AD), and observed an interaction between amyloid beta (Aβ) oligomers and NRXN1/2 that diminished presynaptic organization [51]. Additionally, Aβ oligomers were found to interact specifically with NRXN2α and neuroligin 1 to mediate synapse damage and memory loss in mice [53].

Limitations of this study are the limited sequencing scope of WES, which includes its inability to accurately detect copy number [54, 55] and non-coding variants [56], and the fact that each of the functional prediction tools used relies on their own in-built algorithm, which can produce inaccurate results. Another limitation is that the LNS-1 domain of the protein could not be modelled due to the lack of sequence coverage between the target sequence and homologous template structure. Also, the short simulations times (100ns) used may not allow NRXN2 to sample enough of the protein's energy minima landscape or phase space thereby missing important energetic conformations.

In conclusion, we identified a novel candidate gene, *NRXN2*, for PD thereby potentially implicating synaptic dysfunction in neuronal cell death. Future studies will involve wet-laboratory functional studies in relevant disease models to determine the biological significance of the variant identified.

## Supporting information

**S1 Table. Clinical and demographic information on members of South African family ZA253 with Parkinson's disease.**
(PDF)

**S2 Table. Summary of whole exome sequencing metrics in the three affected individuals.**
(PDF)

**S3 Table. Single nucleotide variants identified in three Parkinson's disease-affected individuals using whole exome sequencing.**
(XLSX)

**S1 Fig. Total energy of the four systems of NRXN2 (WT_noNAG, MUT_noNAG, WT_NAG and MUT_NAG) over 100 ns.**
(PDF)

**S2 Fig. Potential energy of the four systems of NRXN2 (WT_noNAG, MUT_noNAG, WT_NAG and MUT_NAG) over 100 ns.**
(PDF)

**S3 Fig. The average temperature of the four systems of NRXN2 (WT_noNAG, MUT_no-NAG, WT_NAG and MUT_NAG) over 100 ns.**
(PDF)

**S4 Fig. 2D interaction diagram showing polar contacts formed between NRXN2 residues and sugar moiety NAG.** Figure generated using PoseView. Dashed lines show hydrogen bond contacts formed NAG and NRNX2 residues.
(PDF)

**S5 Fig. Total energy of the four systems of the repeat 1 of NRXN2 (WT_noNAG, MUT_no-NAG, WT_NAG and MUT_NAG) over 100 ns.**
(PDF)

**S6 Fig. Potential energy of the four systems of repeat 1 of NRXN2 (WT_noNAG, MUT_no-NAG, WT_NAG and MUT_NAG) over 100 ns.**
(PDF)

**S7 Fig. The average temperature of the four systems of repeat 1 NRXN2 (WT_noNAG, MUT_noNAG, WT_NAG and MUT_NAG) over 100 ns.**
(PDF)

**S8 Fig. RMSD deviation of the backbone atoms for the four systems of repeat 1 NRXN2.** MEAN + STDEV (1.71mm ± 0.52, 2.26mm ± 0.52, 1.27mm ± 0.46 and 2.14mm ± 0.79). Line colours: WT_noNAG = green, MUT_noNAG = light magenta, WT_NAG = red and MUT_NAG = blue.
(PDF)

**S9 Fig. Total energy of the four systems of the repeat 2 of NRNX2 (WT_noNAG, MUT_no-NAG, WT_NAG and MUT_NAG) over 100 ns.**
(PDF)

**S10 Fig. Potential energy of the four systems of repeat 2 of NRNX2 (WT_noNAG, MUT_-noNAG, WT_NAG and MUT_NAG) over 100 ns.**
(PDF)

**S11 Fig. The average temperature of the four systems of repeat 2 NRNX2 (WT_noNAG, MUT_noNAG, WT_NAG and MUT_NAG) over 100 ns.**
(PDF)

**S12 Fig. RMSD deviation of the backbone atoms for the four systems of repeat 2 NRXN2, (1.40mm ± 0.33, 1.98mm ± 0.45, 2.15mm ± 0.51 and 1.29mm ± 0.29).** Line colours: WT_noNAG = green, MUT_noNAG = light magenta, WT_NAG = red and MUT_NAG = blue.
(PDF)

**S1 File. Change in secondary structure simulation movie for wild-type NRNX2 noNAG last 50ns.** Also available at: https://www.dropbox.com/s/ucb8bosmaq2tqw2/WT.mp4?dl=0.
(MP4)

**S2 File. Change in secondary structure simulation movie for mutant NRNX2 noNAG last 50ns.** Also available at: https://www.dropbox.com/s/wkdmvyrvrqdumni/MUT.mp4?dl=0.
(MP4)

## Acknowledgments

We thank the study participants for their participation in and contribution to this study. We also gratefully acknowledge the Western Province Blood Transfusion Service for providing the control samples. We thank Prof. Matthew Farrer and his research group at the Djavad Mowaf-hagian Centre for Brain Health, University of British Columbia, Canada for performing the whole exome sequencing. Prof. Chandra Verma is acknowledged for providing constructive input into the simulation methods and interpretation of the results. We would like to thank Drs Suzanne Lesage and Christelle Tesson for screening for the presence of *NRXN2* p.G849D and *CFAP65* p.T1023A in their exome datasets. We would like to acknowledge the Centre for High Performance Computing (CHPC), Rondebosch, South Africa for allowing us to run our molecular dynamics simulations on their cluster.

## Author Contributions

**Conceptualization:** Ruben Cloete, Soraya Bardien.

**Data curation:** Ruben Cloete.

**Formal analysis:** Boiketlo Sebate, Ruben Cloete.

**Funding acquisition:** Alan Christoffels, Soraya Bardien.

**Investigation:** Boiketlo Sebate, Katelyn Cuttler, Ruben Cloete, Marcell Britz, Alan Christoffels, Monique Williams, Jonathan Carr, Soraya Bardien.

**Methodology:** Boiketlo Sebate, Katelyn Cuttler, Ruben Cloete, Soraya Bardien.

**Resources:** Alan Christoffels, Jonathan Carr, Soraya Bardien.

**Supervision:** Ruben Cloete, Monique Williams, Soraya Bardien.

**Writing – original draft:** Boiketlo Sebate, Ruben Cloete.

**Writing – review & editing:** Boiketlo Sebate, Katelyn Cuttler, Ruben Cloete, Marcell Britz, Alan Christoffels, Monique Williams, Jonathan Carr, Soraya Bardien.

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
