## [Decision Letter · Decision Letter 0]

17 Dec 2020

PONE-D-20-30045

Prioritisation of candidate genes for a South African family with Parkinson’s disease using in-silico tools

PLOS ONE

Dear Dr. Bardien,

Thank you for submitting your manuscript to PLOS ONE. After careful consideration, we feel that it has merit but does not fully meet PLOS ONE’s publication criteria as it currently stands. Therefore, we invite you to submit a revised version of the manuscript that addresses the points raised during the review process.

We look forward to receiving your revised manuscript.

Kind regards,

Yona Levites

Academic Editor

PLOS ONE

Journal Requirements:

Reviewers' comments:

Reviewer's Responses to Questions

**Comments to the Author**

1. Is the manuscript technically sound, and do the data support the conclusions?

Reviewer #1: Yes

Reviewer #2: No

2. Has the statistical analysis been performed appropriately and rigorously? 

Reviewer #1: Yes

Reviewer #2: Yes

3. Have the authors made all data underlying the findings in their manuscript fully available?

Reviewer #1: No

Reviewer #2: No

4. Is the manuscript presented in an intelligible fashion and written in standard English?

Reviewer #1: Yes

Reviewer #2: No

5. Review Comments to the Author

Reviewer #1: In the article entitled “Prioritisation of candidate genes for a South African family with Parkinson’s disease using in-silico tools” the authors describe a unique gene variant in the NRXN2 gene among a very small (n=3) group of Afrikaners who clinically presented as “early onset Parkinson’s cases”. The research team dug deep into the molecular structure of the NRXN2 gene using sophisticated algorithms, bioinformatic and in silico tools to better understand how structural variations of NRXN2 could “potentially” lead to pathogenic variants of the NRXN2 gene. Overall, I think the authors did an exceptional job describing the approach. The only major concern I have with this manuscript is that whole exome sequencing was done on only three subjects, albeit they were the only three that showed such variation. This manuscript would be much stronger if you could repeat this study in a larger cohort. With all the available datasets I would be surprised if you couldn’t find a suitable early onset PD dataset. If this gene variant is only affected in this very small group of people, in the same community, could it be environmentally induced? These kinds of arbitrary questions could be mitigated if you looked in a publicly available dataset. Assuming that the “three affected individuals have the same genetic cause of disease” is not something one should assume. We are not in the business of assuming we are in the business of proving, one way or another. The fact that none of the known PD mutations were found in any of the affected individuals is very interesting, but I would like to know if NRXN2 could be found in those mutation carrying individuals that have already been sequenced. The fact that there is no pathological data confirming disease is always an issue. Although all three “carriers” had PD-like features they were individually unique. I wonder if any of the three cases had overlapping neurodegeneration (e.g. comorbidity)? You have written in the text “Notably, NRXN2 was found to be selectively expressed in the substantia nigra”, but in the discussion you clearly show its expression in other regions and in other diseases. Overall, I think the authors did a nice job, it is an interesting paper and worth exploring further.

Reviewer #2: Bardien et al reported the method to filter the putative pathogenic variant of one autosomal dominant PD family in South Africa. They conducted the analysis and finally regarded NRXN2 as the most putative pathogenic variant.

The paper includes some critical problems;

One is the selection of the putative pathogenic gene. Table 1 seems to be most important data in the study. Among the affected patients, it is CFAP65 that all patients harbored. Thus, the authors should focus on CFAP65, not NRXN2 that partially family members harbored. The prediction tools may indicate the high pathogenicity of NRXN2. However, regretfully, those are not concern to be the reliable. Frequently, the prediction tools yield the unfixed data. Each tools showed the different results. The family study indicated CFAP65 as a putative pathogenic gene. Then, the authors should conduct the functional analysis to confirm the pathogenicity.

Second, the study did not include the frequency analysis for the identified variants. Each race has different frequency of the variants. In a South African population, how is the frequency of CFAP65 or the identified variants. The author should conduct the genetic screening for the cohort of ADPD families or sporadic PD or normal controls. It needs over 200 in each. Because the prevalence is quite low, under 2 or 3%, of the pathogenic genes of familial PD such as SNCA, LRRK2 e.g.

If possible, the more data are need regarding the clinical findings in each patient. For instance, those are brain MRI, CT, response to levodopa, the assessments of cognitive function, or evaluation for psychiatric disorders. Then, make a table to show them. Needless to say, it is also important to evaluate precisely the clinical characters, especially in the patients with young-onset or hereditary PD.

Introduction and discussion is lengthy. They are involved the redundant description. Focus more important findings throughout the text.

6. PLOS authors have the option to publish the peer review history of their article (what does this mean?). If published, this will include your full peer review and any attached files.

Reviewer #1: No

Reviewer #2: No

---

## [Author Response · Author response to Decision Letter 0]

1 Mar 2021

Response to reviewers’ comments

1 March 2021

To: Academic Editor: Yona Levites

PLOS ONE

Dear Dr. Levites

RE: Resubmission of manuscript # PONE-D-20-30045 entitled “Prioritization of candidate genes for a South African family with Parkinson’s disease using in-silico tools” 

We thank the reviewers for their positive feedback and their constructive comments, which have improved the quality of our manuscript. We would also like to thank you and your editorial team for providing us with an opportunity to submit a revised version of this manuscript. Our responses to the reviewers’ comments and questions are provided below. In the marked-up copy of the manuscript, the relevant amended text is shown using track-changes. 

Response to comments on Journal Requirements:

Response: Thank you. We have formatted the document and renamed all files according to PLOS ONE requirements. 

2. We note that you have indicated that data from this study are available upon request. PLOS only allows data to be available upon request if there are legal or ethical restrictions on sharing data publicly. 

Response. As there are no ethical or legal restrictions to sharing of de-identified data, the list of single nucleotide variants identified in the three PD-affected individuals are provided in S3 Table.

Please note, all page and line numbers referred to in the responses below are in the marked-up copy of the manuscript. 

Reviewers' comments:

Reviewer #1:

In the article entitled “Prioritisation of candidate genes for a South African family with Parkinson’s disease using in-silico tools” the authors describe a unique gene variant in the NRXN2 gene among a very small (n=3) group of Afrikaners who clinically presented as “early onset Parkinson’s cases”. The research team dug deep into the molecular structure of the NRXN2 gene using sophisticated algorithms, bioinformatic and in silico tools to better understand how structural variations of NRXN2 could “potentially” lead to pathogenic variants of the NRXN2 gene. Overall, I think the authors did an exceptional job describing the approach. 

1. The only major concern I have with this manuscript is that whole exome sequencing was done on only three subjects, albeit they were the only three that showed such variation. This manuscript would be much stronger if you could repeat this study in a larger cohort. With all the available datasets I would be surprised if you couldn’t find a suitable early onset PD dataset. If this gene variant is only affected in this very small group of people, in the same community, could it be environmentally induced? These kinds of arbitrary questions could be mitigated if you looked in a publicly available dataset. 

Response: Whole exome sequencing was in fact done on 24 individuals from 11 South African families and the NRXN2 variant was only identified in one family, which is the family described in this study. To provide this background and perspective, we have provided the following text in the Materials and Methods section on page 6, lines 128-132. 

“ Initially, we selected 11 multiplex South African families from our PD study collection for WES. These families were selected on the basis of having DNA available of at least two affected second degree relatives, PD had been diagnosed by a neurologist, and at least one individual had to have young-onset PD. WES was performed on a total of 24 affected individuals from these families.”

Additionally, we asked a collaborator with access to WES data of 600 PD families to search their data for the presence of the NRXN2 p.G849D variant. They had sequenced a total of 776 individuals (affected and unaffected) of whom 607 have early onset PD (AAO ≤ 50 years). Notably, the NRXN2 variant was not present in this dataset.

We have inserted the following text on page 18, lines 417-419.

‘The NRXN2 variant was also not present in exomes of 600 probands of French, North African and Turkish origins with predominantly early-onset PD. (Suzanne Lesage, personal communication).’

2. Assuming that the “three affected individuals have the same genetic cause of disease” is not something one should assume. We are not in the business of assuming we are in the business of proving, one way or another. The fact that none of the known PD mutations were found in any of the affected individuals is very interesting, but I would like to know if NRXN2 could be found in those mutation carrying individuals that have already been sequenced. 

Response: Of the 600 families referred to in the response to question 1, 73 have a known pathogenic mutation, and none of them carried the novel NRXN2 variant.

3. The fact that there is no pathological data confirming disease is always an issue. Although all three “carriers” had PD-like features they were individually unique. I wonder if any of the three cases had overlapping neurodegeneration (e.g. comorbidity)? 

Response: These three patients all appear to have young onset PD, with prominent dystonia, resembling similar cases with PRKN and PINK1 mutations. Moreover, all three cases are characterized by a long duration of illness, mild autonomic impairment, and delayed and mild impairment of cognition, with absence of REM sleep behavior disorder. This was confirmed by a movement disorder specialist, Prof. Jonathan Carr. 

This text has been inserted under the Results section on page 14, lines 320-324. 

4. You have written in the text “Notably, NRXN2 was found to be selectively expressed in the substantia nigra”, but in the discussion you clearly show its expression in other regions and in other diseases. 

Response: Thank you for correcting this error. The sentence on page 17, line 393, has now been changed to ‘…highly expressed in the brain including the substantia nigra …’

5. Overall, I think the authors did a nice job, it is an interesting paper and worth exploring further.

Response: We thank the reviewer for this positive comment.

Reviewer #2:

Bardien et al reported the method to filter the putative pathogenic variant of one autosomal dominant PD family in South Africa. They conducted the analysis and finally regarded NRXN2 as the most putative pathogenic variant.

1. The paper includes some critical problems; One is the selection of the putative pathogenic gene. Table 1 seems to be most important data in the study. Among the affected patients, it is CFAP65 that all patients harbored. Thus, the authors should focus on CFAP65, not NRXN2 that partially family members harbored. The prediction tools may indicate the high pathogenicity of NRXN2. However, regretfully, those are not concern to be the reliable. Frequently, the prediction tools yield the unfixed data. Each tools showed the different results. The family study indicated CFAP65 as a putative pathogenic gene. Then, the authors should conduct the functional analysis to confirm the pathogenicity.

Response: We fully agree that the prediction tools can be inaccurate and therefore we chose to use six different tools for our study. We did not prioritize the variant in CFAP65 for the following reasons: 

CFAP65 is involved in ciliac processes in the testis, fallopian tubes, and lungs and is associated with the motile cilia pathway. It functions to regulate spermatogenesis and flagella development. Furthermore, the CFAP65 p.T1023A variant is present in 25 individuals in gnomAD (MAF of 8.88e-5; https://gnomad.broadinstitute.org/variant/2-219886565-T-C?dataset=gnomad_r2_1). For these reasons, the variant in CFAP65 was not prioritized for further study.

2. Second, the study did not include the frequency analysis for the identified variants. Each race has different frequency of the variants. In a South African population, how is the frequency of CFAP65 or the identified variants. The author should conduct the genetic screening for the cohort of ADPD families or sporadic PD or normal controls. It needs over 200 in each. Because the prevalence is quite low, under 2 or 3%, of the pathogenic genes of familial PD such as SNCA, LRRK2 e.g.

Response: For the NRXN2 p.G849D and CFAP65 p.T1023A variants, we have screened additional South African Afrikaner controls, to increase the sample size to >200, as recommended by the reviewer. The following text has been amended on page 17, lines 378-382:

‘Therefore, we screened South African ethnic-matched controls, but none of the variants was present in these individuals. The TEP1 p.Y412C, RFT1 p.A463G and CCNF p.C363S variants were screened in 192 individuals, and the CFAP65 p.T1023A and NRXN2 p.G849D variants were screened in 218 and 216 individuals, respectively.’ 

Also, in the publically available gnomAD (https://gnomad.broadinstitute.org/), the NRXN2 p.G849D variant is not present in this collection of >140,000 unrelated individuals although 2,066 variants in NRXN2 are reported in the database, 785 of these being missense variants. The CFAP65 p.T1023A variant is present in gnomAD at an allele frequency of 25/281,690 (8.88e-5) in individuals of European ancestry (rs748008106). 

Additionally, in the WES data on 600 PD families, referred to in the responses to Reviewer 1, 171 of these families have ADPD and none has the NRXN2 or CFAP65 variants. 

3. If possible, more data are need regarding the clinical findings in each patient. For instance, those are brain MRI, CT, response to levodopa, the assessments of cognitive function, or evaluation for psychiatric disorders. Then, make a table to show them. Needless to say, it is also important to evaluate precisely the clinical characters, especially in the patients with young-onset or hereditary PD.

Response: Thank you for this comment. The required clinical information has been provided in the amended S1 Table. 

4. Introduction and discussion is lengthy. They are involved the redundant description. Focus more on important findings throughout the text.

Response: As recommended, we have shortened the Introduction and Discussion sections of the manuscript. The Introduction has been shortened from almost 3 to 1.5 pages (pages 3-5). The Discussion section has been shortened from 3 to less than 2 pages (pages 23-25).

5. Additionally, Reviewer 2 commented that the manuscript is not written in Standard English. 

Response: The document has now been changed to American English.

---

## [Decision Letter · Decision Letter 1]

16 Mar 2021

Prioritization of candidate genes for a South African family with Parkinson’s disease using in-silico tools

PONE-D-20-30045R1

Dear Dr. Bardien,

We’re pleased to inform you that your manuscript has been judged scientifically suitable for publication and will be formally accepted for publication once it meets all outstanding technical requirements.

Kind regards,

Yona Levites

Academic Editor

PLOS ONE

Additional Editor Comments (optional):

Reviewers' comments:

Reviewer's Responses to Questions

**Comments to the Author**

1. If the authors have adequately addressed your comments raised in a previous round of review and you feel that this manuscript is now acceptable for publication, you may indicate that here to bypass the “Comments to the Author” section, enter your conflict of interest statement in the “Confidential to Editor” section, and submit your "Accept" recommendation.

Reviewer #2: All comments have been addressed

2. Is the manuscript technically sound, and do the data support the conclusions?

Reviewer #2: Yes

3. Has the statistical analysis been performed appropriately and rigorously? 

Reviewer #2: Yes

4. Have the authors made all data underlying the findings in their manuscript fully available?

Reviewer #2: Yes

5. Is the manuscript presented in an intelligible fashion and written in standard English?

Reviewer #2: Yes

6. Review Comments to the Author

Reviewer #2: Reviewer #2:

Bardien et al reported the method to filter the putative pathogenic variant of one autosomal dominant PD family in South Africa. They conducted the analysis and finally regarded NRXN2 as the most putative pathogenic variant.

1. The paper includes some critical problems; One is the selection of the putative pathogenic gene. Table 1 seems to be most important data in the study. Among the affected patients, it is CFAP65 that all patients harbored. Thus, the authors should focus on CFAP65, not NRXN2 that partially family members harbored. The prediction tools may indicate the high pathogenicity of NRXN2. However, regretfully, those are not concern to be the reliable. Frequently, the prediction tools yield the unfixed data. Each tools showed the different results. The family study indicated CFAP65 as a putative pathogenic gene. Then, the authors should conduct the functional analysis to confirm the pathogenicity.

Response: We fully agree that the prediction tools can be inaccurate and therefore we chose to use six different tools for our study. We did not prioritize the variant in CFAP65 for the following reasons:

CFAP65 is involved in ciliac processes in the testis, fallopian tubes, and lungs and is associated with the motile cilia pathway. It functions to regulate spermatogenesis and flagella development. Furthermore, the CFAP65 p.T1023A variant is present in 25 individuals in gnomAD (MAF of 8.88e-5; https://gnomad.broadinstitute.org/variant/2-219886565-T-C?dataset=gnomad_r2_1). For these reasons, the variant in CFAP65 was not prioritized for further study.

>Reviewer 2

I agree with the comments above.

2. Second, the study did not include the frequency analysis for the identified variants. Each race has different frequency of the variants. In a South African population, how is the frequency of CFAP65 or the identified variants. The author should conduct the genetic screening for the cohort of ADPD families or sporadic PD or normal controls. It needs over 200 in each. Because the prevalence is quite low, under 2 or 3%, of the pathogenic genes of familial PD such as SNCA, LRRK2 e.g.

Response: For the NRXN2 p.G849D and CFAP65 p.T1023A variants, we have screened additional South African Afrikaner controls, to increase the sample size to >200, as recommended by the reviewer. The following text has been amended on page 17, lines 378-382:

‘Therefore, we screened South African ethnic-matched controls, but none of the variants was present in these individuals. The TEP1 p.Y412C, RFT1 p.A463G and CCNF p.C363S variants were screened in 192 individuals, and the CFAP65 p.T1023A and NRXN2 p.G849D variants were screened in 218 and 216 individuals, respectively.’

Also, in the publically available gnomAD (https://gnomad.broadinstitute.org/), the NRXN2 p.G849D variant is not present in this collection of >140,000 unrelated individuals although 2,066 variants in NRXN2 are reported in the database, 785 of these being missense variants. The CFAP65 p.T1023A variant is present in gnomAD at an allele frequency of 25/281,690 (8.88e-5) in individuals of European ancestry (rs748008106).

Additionally, in the WES data on 600 PD families, referred to in the responses to Reviewer 1, 171 of these families have ADPD and none has the NRXN2 or CFAP65 variants.

>There are no further comments.

3. If possible, more data are need regarding the clinical findings in each patient. For instance, those are brain MRI, CT, response to levodopa, the assessments of cognitive function, or evaluation for psychiatric disorders. Then, make a table to show them. Needless to say, it is also important to evaluate precisely the clinical characters, especially in the patients with young-onset or hereditary PD.

Response: Thank you for this comment. The required clinical information has been provided in the amended S1 Table.

>There are no further comments.

4. Introduction and discussion is lengthy. They are involved the redundant description. Focus more on important findings throughout the text.

Response: As recommended, we have shortened the Introduction and Discussion sections of the manuscript. The Introduction has been shortened from almost 3 to 1.5 pages (pages 3-5). The Discussion section has been shortened from 3 to less than 2 pages (pages 23-25).

>There are no further comments.

5. Additionally, Reviewer 2 commented that the manuscript is not written in Standard English.

Response: The document has now been changed to American English.

>There are no further comments.

7. PLOS authors have the option to publish the peer review history of their article (what does this mean?). If published, this will include your full peer review and any attached files.

Reviewer #2: No

---

## [Editor Report · Acceptance letter]

18 Mar 2021

PONE-D-20-30045R1 

Prioritization of candidate genes for a South African family with Parkinson’s disease using in-*silico* tools 

Dear Dr. Bardien:

I'm pleased to inform you that your manuscript has been deemed suitable for publication in PLOS ONE. Congratulations! Your manuscript is now with our production department. 

Kind regards, 

on behalf of

Dr. Yona Levites 

Academic Editor

PLOS ONE